# Importance of Skills Development for Ensuring Graduates Employability: The Case of Bangladesh

**Md Jahangir Alam** [1,*], **Keiichi Ogawa** [2] **and Sheikh Rashid Bin Islam** [1]

1   Department of Japanese Studies, University of Dhaka, Dhaka 1000, Bangladesh
2   Graduate School of International Cooperation Studies, Kobe University, Kobe 657-8501, Japan
*   Correspondence: mjalam.jsc@du.ac.bd; Tel.: +880-1716605128

**Abstract:** Graduate employability is a multifaceted concept considering the Sustainable Development Goals. Graduate employability and skills development are also significant determinants for future career success. Graduate employability has seen more sweeping emphasis and concerns in national and global job markets, due to the ever-rising number of unemployed people, which has increased even more due to COVID-19. Due to its importance, this study investigates the current state of skill development initiatives in Bangladesh and the perceptions of university graduates regarding skill development for their future employability. This study uses mixed-method research. Data was collected through surveys and in-depth interviews; various probabilistic and non-probabilistic sample selection methods were used. A total of 437 participants responded to this study. After analysis, the data was shown descriptively. The empirical findings of this study demonstrated that university graduates are well-aware of the skill development requirements for their future employability. However, university graduates face many obstacles in acquiring these necessary skill development opportunities. Therefore, the government and relevant stakeholders must work together to alleviate the obstacles. Furthermore, this study includes recommendations that can assist in developing a model for skill development programs and initiatives in the country for university graduates to ensure their future employability.

**Keywords:** skills development; graduate employability; university; SDGs; COVID-19; Bangladesh

## 1. Introduction

The COVID-19 pandemic struck the entire world by complete and total surprise. As a result of the pandemic, several changes occurred in how the world perceived everything, and various restrictions were put in place to reduce its impact. The COVID-19 pandemic resulted in a significant shift in the way people worked, their working environment, how business and services were conducted, and the competency and skills required for various future employment positions (UNSECO 2020; ILO and World Bank 2021; OECD 2021). Consequently, the advent of COVID-19 showed a significant need for competency and skill development in various sectors for individuals, particularly fresh university graduates, to be employed during and after the pandemic. This had contributed to the already existing unemployment concerns due to a lack of competency and skill development which had resulted in more employees losing their jobs and university students being unable to find employment right after graduating. It is due to the shift of more and more employers towards a hybrid work environment focusing on the 4th Industrial Revolution (4IR) in many parts of the world. This may result in various types of unemployment, including structural unemployment, due to changes in the type of industries and in-demand skills (Blustein et al. 2020; Green 2021; Matli and Ngoepe 2020; Pasara and Dunga 2022; Ferreira et al. 2017; Ingrassia 2003). Despite the importance of competency building and skill development among individuals, many countries worldwide, specifically developing countries, are confronted with the challenge of university graduates lacking skills development and future

employability competency for changing industry demands. The lack of competency and skill development for 4IR and changing industry demand was already an existing problem for graduate unemployment in Bangladesh (Khatun and Saadat 2020). It is primarily due to a lack of social justice, a digital divide, a non-uniform education curriculum with varying levels of education quality, gender disparity, and a lack of government initiatives, among other factors (Uddin 2021; Alam et al. 2022; Parvin and Alam 2016; Duta and Iannelli 2018). As a result, many recent graduates find themselves out of work each year. With the ongoing COVID-19 pandemic, the situation deteriorated further; as more individuals found themselves out of work due to the economic recession and their lack of competency, as well as skills required for future employment. Moreover, the competency and skills needed for employability have also transformed due to the worldwide health crisis (Hossain 2021; Shahriar et al. 2021; Hurrell 2016; Fugate et al. 2004). One might argue that the changing demand of industries for competency and abilities may be due to changes in the labor market and technological innovations, among many other factors. The argument is quite valid; however, the effect of COVID-19 in accelerating these industry changes in many developing countries is often downplayed. During COVID-19, the demand for services and products of various industries, including informal and low technical skill-based, has significantly reduced. The adaptation of technological innovation has increased in many parts of the world during the COVID-19 period. Resulting in a shift in demand for competency and skills to complement this adaptation. Moreover, more and more industries are shifting their focus on 4IR. Hence employment opportunities in many primary and secondary industries are decreasing even faster, where most of Bangladesh's workforce's employment lies. These effects are becoming visible in many parts of the world, including Bangladesh (Abdin 2019; Islam 2020; Schislyaeva and Saychenko 2022). Currently, there is a high emphasis on technical or technological skills, such as information and communicational technology (ICT) skills, as well as various other soft skills by employers in Bangladesh. Most of the in-demand competencies and skills are not possessed by many university graduates due to a lack of initiatives from the tertiary educational institutes of the country (Uddin 2021). Because of this, the future of university graduates in Bangladesh is in jeopardy. The need for skills development for employment in contemporary times and the post-COVID-19 period has become increasingly apparent. Moreover, Bangladesh aspires to meet the United Nations (UN) Sustainable Development Goals (SDGs) by the year 2030 and to achieve the status of a developing country by 2041. Competency and skills development is becoming an increasingly important component of the country's road to becoming sustainable in all areas, including sustainable employability, which can be ensured through graduates' employability (Chowdhury 2022; Munir 2019; Mehrabani and Mohamad 2015). Despite the well-aware gap in skill development needed for university graduates to ensure their future employment in the post-COVID-19 period, significantly less is known regarding the perception and knowledge of university students on the matter; which is a typical case in many parts of the world as well (Gedye and Beaumont 2018). Moreover, according to the theory of employability and human capital theory, skill development plays a vital part in an individual's employability and career development, which may contribute to the nation's overall development as the country will possess more skilled human resources (Knight and Yorke 2003; Nafukho et al. 2004). Therefore, it can be said that skill development is crucial for university graduates, not only for their employability and progress; but also, for their contribution to the nation's development. Hence, Bangladesh must move quickly and understand the perception and needs of university graduates to ensure their skills development and future employment. In contrast to Bangladesh, many international counterparts have ensured its graduates have the requisite competency and skills for future and post-COVID-19 employment. This is enacted through the various existing practices and systems for competency and skills development initiatives that have been put in place by those countries to ensure the sustainable employability of their graduates. One such international counterpart is Japan, where it has been possible to accomplish near complete graduate employability through skill development through career counseling,

career education, the prevalence of vocational and practical education, and the close relationship between educational institutions, among many other initiatives (Woolgar et al. 2008; Woolgar 2007; OECD 2020). Hence, given the importance of competency and skills development for university graduates' future employability, this study attempts to illustrate the contemporary situation of Bangladesh's skills development programs for university graduates; analyze the initiatives taken by the government and stakeholders to implement skills development programs for university graduates in order to ensure their future employability. Finally, it concludes by providing recommendations that may benefit Bangladesh in assuring skill development for university graduates and their future employability in post-COVID-19 periods.

## 2. Theoretical Framework

Within the field of sociology of education, diverse theories have served as the theoretical framework for the graduates' employability debate. One such theoretical paradigm has been the consensus theory on graduate employability, which has been the focus of much previous research. The consensus theory of employability states that enhancing graduates' employability and advancing their careers requires improving their human capital, specifically their skill development (Selvadurai et al. 2012).

In addition, the human development theory and the human capital theory come to the forefront whenever employability is considered. In the context of human development theory and the human capital theory, human development and human capital are terms used to describe the quality of graduates, which consists of the level of skill and knowledge that graduates bring to their employment. Human development and human capital are therefore equated to the definition of employability skills, which refers to graduates' knowledge and skills (Jonck 2014; Jonck and Van Der Walt 2015).

Therefore, based on the theoretical and empirical background of employability skills, this study believes that skills development will positively affect the employability indicators of graduates, which in turn will lead to their future employability. Furthermore, the study will make a theoretical contribution by outlining graduates' understanding of the importance of skill development for future employability.

## 3. Background and Literature Review

### 3.1. Effects of COVID-19 on Unemployment in Bangladesh

Hossain (2021) mentions that Bangladesh will already have a total workforce of 70 million people by 2020, where the agriculture and service sectors account for 79 percent of overall employment, with the remainder concentrated in the industrial sector. The service sector was the most dominant in that same year, accounting for around 40 percent of total employment, and is expected to expand even further in the coming years. Moreover, Kumar and Pinky (2021) states that, as a result of the ongoing COVID-19 crisis that has affected the entire world, similarly, Bangladesh has experienced a tremendous social and economic crisis. Additionally, with the decreasing demand for various commodities and services, the pandemic had other observed effects. The majority of this was due to the different control measures implemented by the government for various sectors of the economy, which had a negative impact on the employment and livelihood of the citizens of the country, expressed Genoni et al. (2020). The COVID-19 outbreak, associated healthcare costs, and ensuing interruptions are expected to exacerbate the unfavorable effects on employment even more in the coming months. Trading Economics (2020) reports that the official unemployment rate in Bangladesh was around 4 percent when the pandemic first hit the country in 2020; moreover, the unemployment rate for an educated workforce was increasing to nearly 2.2 million each year as well. This increased unemployment rate can be considered to be due to the accelerated shift towards 4IR by industries and the economic instability faced by many primary and secondary industries due to increased technological innovation adaptation during COVID-19 (Islam 2020; Khatun and Saadat 2020; Abdin 2019). Moreover, Bangladesh suffered from a stable unemployment rate of 4.4 percent from 2013

to 2019, which increased to 5.4 percent in 2020 and 5.2 percent in 2021 due to the COVID-19 pandemic (The World Bank 2022). This increase is illustrated in Figure 1.

**Figure 1.** Total unemployment rate of Bangladesh from 2013 to 2021. Source: Created by the authors, based on the data from The World Bank (2022).

As per the Policy Research Institute, due to the COVID-19 health crisis and associated government initiatives to limit the effects of the pandemic, the country's unemployment situation deteriorated much sooner rather than predicted during pre-COVID-19 periods (PRI 2020). According to further research by Ali and Bhuiyan (2020), the overall unemployment situation in the country will worsen in the years to come due to the COVID-19 health crisis, with a significant number of individuals losing their jobs and having significantly higher chances of losing their employment in the future. As more and more industries shift their focus and demand for competencies and skills.

*3.2. Need of Skills Development for Future Employability*

Like no other decade in modern history, the COVID-19 pandemic has fundamentally altered how the world perceives normalcy in just a few months. It has already influenced how businesses and services are perceived around the world, which will continue to change in the future (Buheji 2020; OECD 2019). Meister (2020) expressed that the world must reshape itself so that it conforms to the post-COVID-19 status quo as it endeavors to resume its activity toward enhanced effectiveness, advancement, and sustainability. Consequently, most prominent organizations in Bangladesh would devote their resources to training their workforce again or recruiting according to the new typical standards emphasizing 4IR, resulting in fresh university graduates needing to require additional competency and skills development to find employment in the post-COVID-19 period. Tertiary educational institutes and similar establishments would play a vital role in the post-COVID-19 new normal. Subsequently, it should be expected that developing employability competency through skills development will be at the top of every nation's priority list in the post-COVID-19 era; if the nation wishes to become healthy, more competent, and sustainable, instead of producing highly educated graduate students from tertiary educational institutes with outstanding academic performance alone. The primary goal will be to produce competitive university graduates from tertiary educational institutes who can overcome unexpected

and anticipated obstacles, and turn them into a source of development and differentiation for their organizations through the utilization of the skills they have gained through education and training. As a result, in post-COVID-19 periods, academic competence alone will no longer be sufficient in the eyes of employers for employing university graduates, and without proper competency and skills development, university graduates will fail to secure future employment in post-COVID periods (Buheji and Buheji 2020; Paulos et al. 2021). However, multiple works over the year by Shohel et al. (2021), Monem and Baniamin (2010), and Hossain et al. (2016) have shown that the higher educational (HE) institutes in the country are still following outdated curriculums and have not adopted the use of ICT in education. Moreover, these HE institutes play close to no role in developing the skills or competency of graduates for their future employability. This lack of HE institutes in the country became more evident during COVID-19. In comparison, international HE institutes are adopting the use of new technology, innovation, and policies to address these issues (Clarke 2018). Subsequently, research by Nusrat and Sultana (2019), Shahriar et al. (2021), Milon et al. (2021), and Chowdhury and Miah (2016) have pointed out that employers have shown a strong emphasis on the skills and competency of graduates when hiring. This emphasis on skill development and competency has become even more evident by the employers in the country during COVID-19. Hence, skills development for future employability is becoming more and more critical.

### 3.3. Skills Development for Achieving SDGs

The SDGs, or the 2030 Agenda for Sustainable Development, are an action plan adopted by all the 193 states that are members of the UN. The SDGs are a set of 17 goals intended to eradicate poverty and bring about crucial and revolutionary changes that are immediately necessary to create a sustainable and resilient future for all (UN 2016). According to a widely held opinion, it is vital to break the negative spiral of inadequate skills, poor productivity, and inequality to promote equitable growth in the economy and jobs for everybody, which will ensure sustainability. Quality education is not just a goal in and of itself; it is also a way of obtaining respectable employment, particularly for young people. According to studies, continuous skills development is essential to keep up with the ever-changing demands of a competitive labor market (ILO 2016). As a result, skills development may be recognized as a prerequisite for sustainability, and it is a critical component of accomplishing the SDGs or the 2030 Agenda for Sustainable Development. Predominantly, this could be attained through SDG 4, 'Quality Education,' and SDG 8, 'Job Creation, Decent Employment and Retraining;' this, in turn, will contribute to ensuring sustainability (King 2016).

The unemployment rate of Bangladesh is severely affected by the COVID-19 health crisis. It has also been pushed up due to the lack of competency and skills needed for future employability in the post-COVID-19 period by university graduates. Furthermore, skills development's importance for assisting SDG achievement is quite evident. As the globe transitions into a post-COVID-19 period, Bangladesh must act to ensure future-ready skills development and graduate employability for its sustainable development. However, considering the significance of skill development of university graduates in Bangladesh for employability and sustainability, there is a vast knowledge gap on how to achieve it. This is due to close to little previous work on the topic and a lack of understanding of the perception and needs of university graduates. Hence, this study aims to fill the knowledge gap by understanding university graduates' perceptions of skill development and future employability. The study hopes to fulfill its objective of filling the knowledge gap by responding to the following research questions:

RQ1. What is the current state of skill sets possessed by the graduates, and if they consider it enough for future employability? Moreover, if they face any barriers in accessing skill development programs?

RQ2. Is there any existing skill development program offered by the government or educational institutes?

RQ3. Are the graduates aware of the need for skills development for future employability?

## 4. Materials and Methods

### 4.1. Sample Size

The total sample size for this study is 447. All the participants are university students from Bangladesh and are currently enrolled in a full-time educational program at a local tertiary educational institute and are expected to graduate soon. The participants are selected from 46 institutes and over 72 faculties. The participants come from different locations within the nation and various socio-economic situations, which contributes to ensuring the proper presentation of the population. There were 246 males and 172 females among the 437 participants (age range = 18–28; mean age = 22 years; SD = 1.97) that took part from Bangladesh. Among the participants, 79.9 percent ($n = 349$) are currently pursuing their undergraduate, 19 percent ($n = 83$) are pursuing their masters, and only 1.1 percent ($n = 5$) are pursuing their doctoral studies. Moreover, all the participants further informed that they would seek employment right after graduation. However, when asked if they were involved in part-time employment, 80.1 percent ($n = 358$) of them said they were not in employment, while 19.2 ($n = 84$) percent stated they were employed part-time. Out of the 447 participants, 20 students were randomly selected for an in-depth interview (IDI). The students who participated in the IDI consisted of twelve males and eight females; moreover, 45 percent ($n = 9$) of them were enrolled in their master's program, while the rest 65 percent ($n = 11$), were pursuing their doctoral studies. The characteristics of the total sample used in the study are shown in Table 1.

**Table 1.** Sample characteristics.

| Characteristics | | Frequency | Percentage |
|---|---|---|---|
| Sex | Male | 271 | 60.6 |
| | Female | 176 | 39.4 |
| Enrolled Education Program | Undergraduate | 349 | 78.1 |
| | Master's | 83 | 18.6 |
| | Doctoral | 15 | 3.3 |
| Current Employment Status | Not in Employment | 358 | 80.1 |
| | Part-Time | 89 | 19.9 |

### 4.2. Research Design and Instrument

A mixed-method is used for this study, as well as abduction reasoning. A mixed research methodology is selected for quantifying the responses of the graduates on the skill development for future employability; this will allow the analysis of the quantitative data and help understand how the relationship between skill development and future employability through data analysis; while the qualitative data will allow a deeper understanding of the perception and views of the graduates regarding skill development and future employability. Subsequently, the methodology for this study is designed in accordance with similar studies conducted by Cruz-Sandoval et al. (2022), and Pham (2022). This study acquired data from January to February 2022 and performed a member check in May 2020. The data in this research is collected through an online self-completion survey and through IDI. The survey included a 22-item standardized questionnaire. Of the 22 questions, four questions included a Likert scale for the quantitative method and four questions included multiple-choice and checkbox for checking participant's frequency on various questions relating to skill development and future employability as a qualitative

method. The questions asked in the survey used for data analysis are shown in Table 2. The selected online service for the survey questionnaire is Google Forms. The distribution to the participants was conducted through various social networking sites and academic emails. A non-probabilistic snowball sampling and convenient sampling method were used for sample selection. All the participants who responded to the survey are from Bangladesh. For the IDI, only 20 interviews were conducted as a data saturation point was achieved. The interview was conducted using a semi-structured eight-item standardized questionnaire. A probabilistic simple random sample selection method was used for the IDI. The document analysis technique was also used to collect secondary data, including newly published educational materials and reports. Research ethics were ensured throughout the data collection process. The participants provided their consent for participation and were well aware of the objective of the research.

**Table 2.** Questions asked in the survey used for data analysis.

| Item | Survey Question | Type |
|:---:|:---|:---:|
| 1 | Evaluate your current skillset for future employability based on your hard skills. | Quantitative |
| 2 | Evaluate your current skillset for future employability based on your soft skills. | Quantitative |
| 3 | Evaluate your current skillset for future employability based on your technological skills. | Quantitative |
| 4 | Evaluate your employability chances based on your existing skill set. | Quantitative |
| 5 | Do you think there is a mismatch between the current skill sets graduates to possess and the required skill sets for employment? | Qualitative |
| 6 | Do you consider the higher education institutes were doing enough to ensure sustainable skills development for university students' employability? | Qualitative |
| 7 | Select the employment sectors you would prioritize the most while starting your careers. | Qualitative |
| 8 | Did you have one or multiple barriers to accessing skills development programs? | Qualitative |

*4.3. Analysis Process*

The data collection and processing are conducted in accordance with the recommendations by Palinkas et al. (2015) and Jansen (2010) for mixed-method research. The data acquired from the survey has been managed, cleansed, and transformed. The assistance of the statistic computer software for social sciences IBM SPSS version 25 and Microsoft Office Excel 2019 were used for analysis. After analysis, the empirical findings have been represented descriptively and graphically. The data collected from IDI has been used for identifying and assisting in proposing the lessons and recommendations which can be learned regarding the study's theme. From Table 3, it can be seen that the Cronbach's Alpha value for quantitative items is 0.983, indicating that the questionnaire's reliability is excellent and acceptable for this study.

**Table 3.** Reliability of the quantitative questions asked in the survey.

| No. | Question | Cronbach's Alpha |
|:---:|:---|:---:|
| 1 | Evaluate your current skillset for future employability based on your hard skills. | |
| 2 | Evaluate your current skillset for future employability based on your soft skills. | |
| 3 | Evaluate your current skillset for future employability based on your technological skills. | 0.983 |
| 4 | Evaluate your employability chances based on your existing skill set. | |

## 5. Results

### 5.1. Need for Further Skills Development for Graduate Employability

The participants were asked to evaluate their current skillset for future employability based on three specific skill sets. These three skills are often mentioned in regard to youth employability according to reports published by BRAC, Bangladesh, and International Labour Organization (ILO) (Islam et al. 2020; ILO 2020); and these three skill sets are hard skills, soft skills, and technological skills, which can be considered as a branch of technical skills. Hard skills can be associated with foreign language skills or professional certification; soft skills can be associated with communication, leadership, and time management; and technological skills can be associated with the capability of using technological devices and software. The frequency for this question was ranked 1 for 'not enough for employment,' 2 for 'needs more development for employment,' and 3 for 'enough for employment.' Table 4 reveals that the university students consider that they do not have enough hard skills (M = 2.79; SD = 1.49), the soft skills (M = 2.59; SD = 1.16), as well as technological skills (M = 2.53; SD = 1.15) for future employability and think they require skill devolvement for future employability.

**Table 4.** Mean score of frequency of evaluation of current skills set for future employability.

| Name of Skill Sets | Hard Skills | Soft Skills | Technological Skills |
|:---:|:---:|:---:|:---:|
| N | 437 | 437 | 437 |
| Mean | 2.79 | 2.59 | 2.53 |
| Standard Deviation | 1.49 | 1.16 | 1.15 |

Table 5 lists the response of the 437 participants to this question. Most of the participants replied that they did not have enough skills to ensure future employment; 31.0 percent (*n* = 135), 34.5 percent (*n* = 151), and 33.0 percent (*n* = 144) of participants selected 'needs more development for employment' for hard skills, soft skills, technological skills, respectively. In contrast, 20.0 percent (*n* = 87) of participants felt they had enough hard skills for future employment, which was the most significant response for the 'more than enough for employment' value among the three skills; while the response for this value for the other two skills, soft and technological skills, was the lowest at 4.9 percent (*n* = 21) and 5.0 percent (22), respectively. Around 24.0 percent (*n* = 105) of participants responded that they did not have enough hard skills for employment, again the most extensive response for the 'do not have the skills for employment' section.

**Table 5.** Frequency of evaluation of current skills set for future employability.

| Level of Evaluation | Hard Skills | Soft Skills | Technological Skills |
|---|---|---|---|
| N | 437 (100.0%) | 437 (100.0%) | 437 (100.0%) |
| Do not have the skills for employment | 105 (24.0%) | 84 (19.2%) | 92 (21.0%) |
| Not enough for employment | 135 (31.0%) | 151 (34.5%) | 144 (33.0%) |
| Needs more development for employment | 66 (15.0%) | 84 (19.2%) | 100 (22.9%) |
| Barely enough for employment | 44 (10.0%) | 97 (22.2%) | 79 (18.1%) |
| More than enough for employment | 87 (20.0%) | 21 (4.9%) | 22 (5.0%) |

The participants were further asked to evaluate their employability chances based on existing skills set on a Likert scale containing 5 points. The frequency was ranked as 1 for 'very less chances for employment,' 2 for 'low chances for employment,' 3 for 'average chances for employment,' 4 for 'high chances for employment,' and 5 for 'very high chances for employment.' From Table 6, it can be observed that most of the graduates think they have a low level (M = 2.76; SD = 1.21) of the possibility of securing employment based on their current skill sets.

**Table 6.** Mean score of frequency of employability chances based on existing skills set.

| | N | Mean | Standard Deviation |
|---|---|---|---|
| **Employability Chances** | 437 | 2.81 | 1.78 |

From Table 7, it can be seen that most participants (*n* = 437) for this question evaluated themselves to have 'average chances for employment,' which had a selection rate of 27.7 percent (*n* = 121). Whereas 'very high chances for employment' was selected as the lowest among the participants and had a section rate of 5.9 percent (*n* = 26). Moreover, from the data, it can be revealed that 68.2 percent of university students consider that their current skill set may not provide them employment.

**Table 7.** Frequency of employability chances based on existing skills set.

| Employability Chances | Frequency | Percentage |
|---|---|---|
| N | 437 | 100.0 |
| Very less chances for employment | 90 | 20.5 |
| Low chances for employment | 87 | 20.0 |
| Average chances for employment | 121 | 27.7 |
| High chances for employment | 113 | 25.9 |
| Very high chances for employment | 26 | 5.9 |

To access the correlation between the self-evaluation of the graduates on their 'hard skills,' 'soft skills,' 'technological skills,' and their 'employability chances,' a Spearman's rank correlation coefficient test and gamma test have been carried out. Table 8 shows that employability chances have a Spearman's Rho value of 0.943, 0.943, and 0.940 with 'hard skills,' 'soft skills,' and 'technological skills', respectively. Moreover, it has a Gamma score of 1.000 with 'hard skills,' 'soft skills,' and 'technological skills' as well. These scores can

be considered a near-perfect association of ranks between the variables. This means the variables are positively and highly correlated. This positive correlation is also illustrated in Figure 2.

**Table 8.** Spearman's Rho and Gamma value of variables.

| Variables | Employability Chances | |
| --- | --- | --- |
| | Spearman's Rho | Gamma |
| Hard Skills | 0.943 | 1.000 |
| Soft Skills | 0.943 | 1.000 |
| Technological Skills | 0.940 | 1.000 |

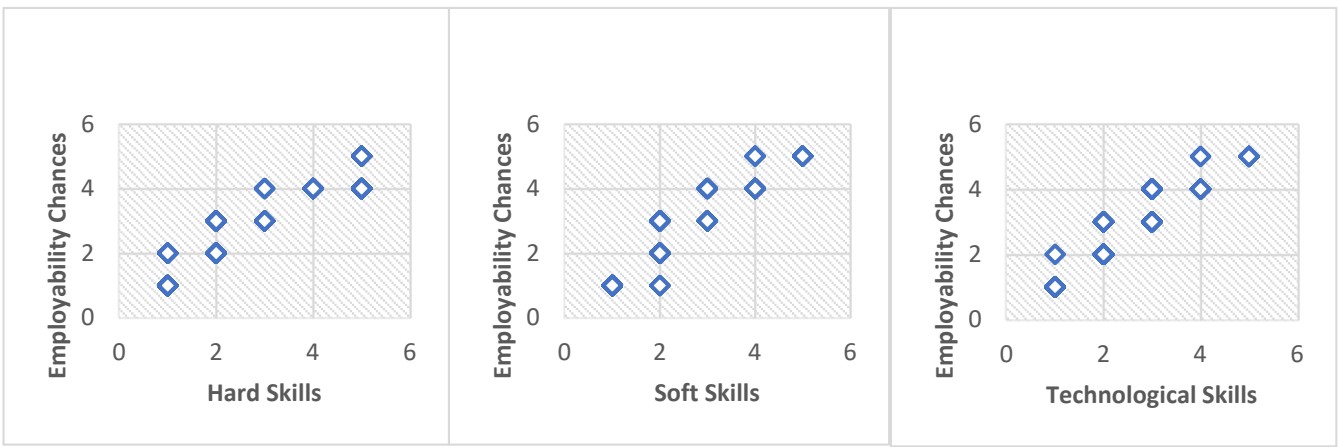

**Figure 2.** Scatterplot showing relationship between the self-evaluation of the graduates on their 'hard skills,' 'soft skills,' 'techno-logical skills,' and their 'employability chances'.

*5.2. Lack of Initiative for Skills Development Program*

Furthermore, the participants were asked whether they thought there was a mismatch between the current skill sets graduates to possess and the required skill sets for employment; and whether they considered the higher education institutes were doing enough to ensure skills development for university graduates future employability. The answer to both questions is graphically presented in Figures 3 and 4, respectively. For the first question, 81 percent ($n = 355$) of participants responded 'yes,' while 19 percent ($n = 82$) responded 'no.' As for the second question, 73 percent ($n = 318$) responded 'no,' In comparison, 27 percent ($n = 119$) responded 'yes.' Hence, it can be seen from the data that the university students understand there is a mismatch between the skills they possess right now and the skills set they require to be employed in the coming days and the post-COVID-19 period. Furthermore, despite the awareness of the university students, there has been very limited or no initiative from the tertiary education institutes and related establishments to ensure post-COVID-19 skills development for the university students and to ensure future employability.

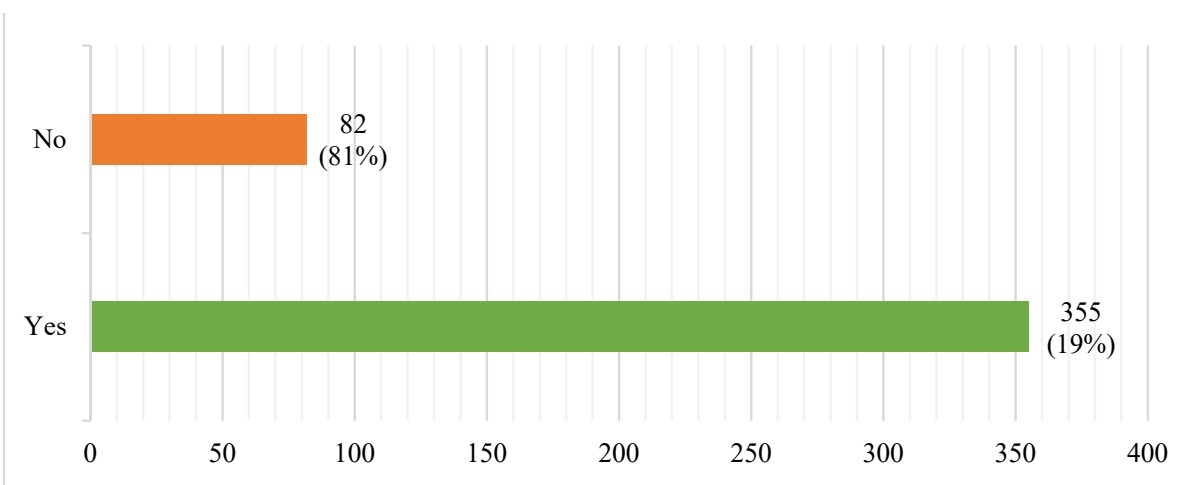

**Figure 3.** Frequency for the response for 'is there existing mismatch between the current and require skills sets needed for graduate employability.'.

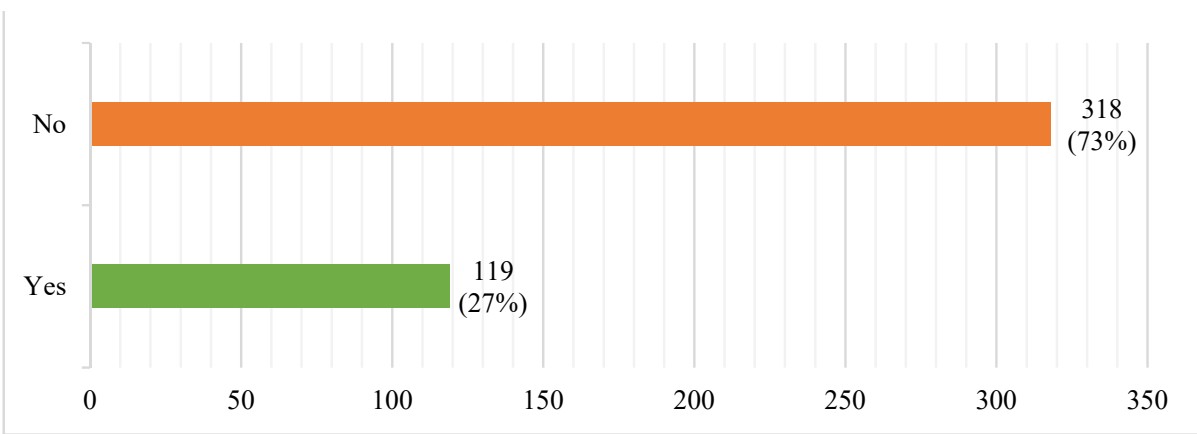

**Figure 4.** Frequency for 'was any initiative taken by tertiary education institutes for graduates' skills development for their employability'.

*5.3. Graduates' Tendency towards Government Sector Jobs*

The 437 participants were asked to select the employment sectors they would prioritize the most while starting their careers. They prioritized the sectors based on first, second, and third choices. Where 'first choice' is the highest and 'third choice' is the lowest. The results are shown in Table 9 showcase many participants who prioritized government services as their preferred employment sector for starting their careers. 25.9 percent ($n$ = 113), 24.2 percent ($n$ = 106) and 14.9 percent ($n$ = 65) graduates selected the government sector as their 1st choice, 2nd choice and 3rd choice. Whereas the education sector came out as second most preferred with 22.9 percent ($n$ = 100), 12.8 percent ($n$ = 56), and 11.4 percent ($n$ = 50) as first choice, second choice, and third choice, respectively. At the same time, the construction sector was the least preferred by the graduates with 0 percent ($n$ = 0), 0.4 percent ($n$ = 2), and 1.4 percent ($n$ = 6) as first choice, second choice, and third choice, respectively.

**Table 9.** Graduates Prioritization of Employment Sector.

| Employment Sector | 1st Choice | | 2nd Choice | | 3rd Choice | |
|---|---|---|---|---|---|---|
| | Frequency | Percentage | Frequency | Percentage | Frequency | Percentage |
| N | 437 | 100.0 | 437 | 100.0 | 437 | 100.0 |
| Financial | 71 | 16.2 | 98 | 22.4 | 51 | 11.7 |
| Government Service | 113 | 25.9 | 106 | 24.2 | 65 | 14.9 |
| Manufacturing | 16 | 3.7 | 30 | 6.9 | 77 | 17.6 |
| Research & Development | 75 | 17.2 | 61 | 14.0 | 45 | 10.3 |
| Construction | 0 | 0.0 | 2 | 0.4 | 6 | 1.4 |
| Education | 100 | 22.9 | 56 | 12.8 | 50 | 11.4 |
| Private | 31 | 7.1 | 44 | 10.1 | 42 | 9.6 |
| Entrepreneurship | 27 | 6.2 | 34 | 7.8 | 76 | 17.4 |
| Others | 4 | 0.8 | 6 | 1.4 | 25 | 5.7 |

University graduates often lack the opportunities and required skillsets to get employed elsewhere. Hence, they tend to lean towards the civil service examination, which only considers one's rote learning abilities for employment, not their real-life or employment-related skills. There is also a limited number of employment opportunities in the government sector, therefore resulting in more competition. As more university students target government sector jobs, they focus less on their skills development and more on rote learning specific to civil service examination. Hence, it led to a lack of skills development, resulting in unemployment and a lack of sustainable employment in contemporary times, which may worsen in the post-COVID-19 period.

*5.4. Barriers for Skills Development Program*

When asked if the participants had one or multiple barriers to skills development, only 55 out of the total participants said 'no,' and 227 participants said they had 'financial limitations.' About 126 participants said they had 'technological limitations,' such as unavailability of computer or internet connection, and 225 participants said they had 'knowledge limitations,' such as unaware of available skills development opportunities. Only 2.7 percent (*n* = 12) said they had 'other' existing barriers which have limited their access to skills development. The responses of the participants are shown in Table 10.

**Table 10.** Barriers to involvement in skills development.

| Barriers to Involvement in Skills Development | Frequency | Percentage |
|---|---|---|
| N | 437 | 100.0 |
| None | 55 | 12.6 |
| Financial Limitations | 227 | 51.9 |
| Technological Limitations | 126 | 28.8 |
| Knowledge Limitations | 225 | 51.5 |
| Others | 12 | 2.7 |

The multiple barriers that university students in Bangladesh must overcome considerably lower their chances of gaining access to skills development, reducing their prospects of gaining sustainable employment in the long run. After COVID-19, the ramifications of this will be multiplied since there will be a strong emphasis placed on a variety of skills for future employability.

*5.5. Lacks in Skill Development Initiatives*

The participants who partook in the interview were asked questions regarding the skill development projects in Bangladesh and their lacking. From analyzing the interviews, it can be noticed that all 20 participants shared similar viewpoints. The participants expressed that there was a lack of government and stakeholder initiatives in Bangladesh, and the quality of education in tertiary educational institutes for future employability was considerably low. Furthermore, they stated a lessened prevalence and stigmatization of vocational education in Bangladesh. In comparison, many developed countries consider it quite popular and see it as an alternative route for employment. The lack of academia-industry relations and the absence of career guidance and skill development initiatives by educational institutes were also mentioned as causes of the lack of university students' future employability by the participants. They also emphasized the prevalent trend of graduates focusing on landing employment in the civil sector, which must be reversed. Finally, they stated that the university students must develop their skills or else they might find it difficult to find sustainable employment in the post-COVID-19 period. However, this can be very difficult for them unless the government and stakeholders assist the student in doing so.

During the interview, one of the participants expressed the following regarding skill development initiatives and their lacking in Bangladesh, "*The growing need for skill development for employment is quite evident. More and more of our seniors who have graduated in the last few years have had various difficulties ensuring employment right after graduation. Many of my peers and I also fear that we might not find employment right after graduation . . . Despite all of us knowing the need for skill development for future employment, there is little we can do. There are barely any initiatives by our university, nor are there any by the government. The available skill development programs are offered by private organizations and often are very expensive, which many of us cannot afford . . . Hence, without the assistance of the government, this plight of the graduates cannot be changed and may worsen in the future*" (Participant G).

Moreover, the participants also stressed that their career aspirations often change due to their lack of skills and competency. One such participant expressed, "*most of the graduates, including myself, want a job with a high salary and job security. Unfortunately, most jobs offering these benefits usually ask during the interview if we possess specific additional skills specific to the job or not. People like me who do not have those skills are often not employed. This is the case for many, which have led to go for jobs which are easier to get, an example being the government jobs*" (Participant E).

## 6. Discussion and Recommendations

Several studies and reports have been undertaken and published in the past few years regarding the growing unemployment of university graduates in Bangladesh due to a lack of skills and competencies. These studies also indicated a shortage of university graduates with the necessary skillsets for these positions despite enough job openings. As a result, employees' salaries, opportunities, and job security-related risks have increased as employers claim they hire graduates with less than the requisite competency and skillsets (Khatun et al. 2022; Chisty et al. 2007; Rahman et al. 2021). None of the studies and reports have addressed why this lack of competency and skill development occurs among graduates. From the results obtained in this study, there is a positive correlation between the self-evaluation of the graduates on their 'hard skills,' 'soft skills,' 'technological skills,' and their 'employability chances.' This means that the graduates are well aware that their future employability chances will be reduced if they do not develop their skills. Additionally, it is found that university graduates have a good idea of what type of hard, soft, and technical skills they possess and might need to develop even further. Furthermore, they have also realized that there is a mismatch between the skills they possess and the skills they might require for employment soon. These results suggest that there is no lack of awareness among university graduates regarding what needs to be done for their employability.

The university graduates further have stated that a vast majority consider themselves not to have the competency and skill sets needed for employment. Furthermore, they also indicated there are significantly fewer interventions or initiatives taken by tertiary educational institutes and relative establishments to ensure their skill development for future employability. The lack of initiatives of tertiary educational institutes and other organizations for introducing skill development programs due to various impediments in Bangladesh is well reported by many (Newaz et al. 2013; Shahadat et al. 2012). As the university graduates do not receive any skill development support or guidance from their respective institutes, they have to take it upon themselves to develop their skills. However, as seen from the results, they are confronted by numerous obstacles. These impediments stem from Bangladesh's current socioeconomic problems, which include a lack of social justice, a digital divide, a non-uniform education curriculum with varied levels of education quality, and a lack of government initiatives, among other factors (Uddin 2021; Alam et al. 2022). In addition, there are hurdles to participation in skill development programs and a lack of support from tertiary educational institutes. From the interviews, it is founded that most of the graduates' career aspiration is dominated by ambitions of a high salary and job security. However, employers catering to these aspirations often requires specific skills, which are often not possessed by the graduates; hence their aspirations are often changed later on. The results show that these university graduates' career aspiration is later motivated by easier employability, and they gravitate toward employment areas that need little to no skill development, particularly the government sector. However, these industries have fewer job openings compared to the number of university graduates who apply, resulting in increased unemployment and further lack of skill development. As a result of the findings of this study, it can be noted that university graduates are aware of the skills they will need for employability during and after the COVID-19 pandemic and are willing to improve those skills. They are, however, up against several obstacles. As a result, it can be concluded that university students' lack of skill development for future employability is not attributable to their ignorance. Instead, it is owing to their socio-economic conditions as well as the government's, tertiary educational institutes, and other stakeholders lack of initiative and action.

Compared with Bangladesh, many international counterparts are faring outlying better in terms of skills development for graduates, which in turn has ensured their future employability. One such country is Japan, where around 96 percent of recent graduates have found employment by the end of 2021 (The Japan Times 2021). This accomplishment was only feasible because of Japan's established procedures and system for assisting graduates and ensuring that they are well aware of the skills and experience required for employability in contemporary times and beyond (Saito and Pham 2019, 2021; Ito 2014; Pazyura 2017). Career counseling, career education, the prevalence of vocational and practical education, and the close relationship between educational institutions and employers are just a few of the many existing practices and processes that are well developed in Japan and contribute to graduates obtaining future employment through skills development in Japan (Woolgar et al. 2008; Woolgar 2007; OECD 2020).

*6.1. Recommendations*

Moreover, from the results obtained from the IDI combined with the authors' knowledge regarding the issue, these are the following recommendations that can be made for Bangladesh to ensure skills development for university graduates and their future employability:

First, the government must establish detailed guidelines for tertiary and upper secondary educational institutions to provide their students with adequate career counseling or guidance. This will assist the students in deciding which career they want to pursue and what skills are required to do as well as the skills and expertise that are in high demand after graduation for future employability.

Second, the government must take the proactive initiative to ensure skill development programs for students. These skill development programs must be inexpensive for the students; moreover, it should be offered within the educational institutions for easier accessibility for the student as well.

Third, establishing solid cooperation between tertiary educational institutes and employers to understand industry demands and provide graduates with internship and active learning opportunities. This will, in turn, that skill development programs are offered to address the needs of the employers while assuring graduates' employment readiness through practical experience.

Fourth, the quality of technical and vocational education and training (TVET) must be increased. It must be made more accessible and affordable by introducing it as a part of the educational curriculums for many educational institutes. Furthermore, the government must provide incentives for creating more employment in this sector.

Fifth, reducing graduates' dependency on government or civil sector employment by generating and regulating more attractive employment opportunities in the private sector through a public-private partnership. Moreover, the employment opportunities in the civil sector could also be increased.

Finally, developing training and inclusive educational faculties will assure quality education and institutional globalization to ensure that university students get the education and skills needed for future employability both within their own country and internationally.

By implementing these recommendations and initiatives, Bangladesh may promote and increase skills development in the country for graduates, thereby ensuring their future employability in contemporary times and the post-COVID-19 period, assisting them to achieve SDGs by 2030.

*6.2. Limitations*

This study is not able to outline which skills might be required for this future employability. Moreover, the gap between the perception and needs of university students and employers has also not been identified. Therefore, limitations remain in the study. Hence, this study will serve as a starting point, and further investigation will be required.

## 7. Conclusions

From the obtained results, the significance of skill development for ensuring the future employability of university graduates following the conclusion of COVID-19 is well evident. The university students in Bangladesh are aware of this fact as well. However, the graduates cannot acquire the requisite skill sets due to a scarcity of or restricted possibilities. As a result, the government, tertiary educational institutes, and stakeholders should proactively and effectively examine the exceptional circumstances and prioritize public-private partnerships to achieve success. Additionally, a capable skills development model for Bangladeshi graduates is required to address the country's unemployment crisis in the following years. The authors propose that this can be manifested through a national skill development framework, which will serve as a guideline for the government to take initiatives that will outline the role and duties of educational institutes and stakeholders for the future employability of graduates through skill development.

Furthermore, because skill development can be considered a necessary pre-condition for sustainability in all areas of a country and SDGs achievement, Bangladesh must accord it the highest priority possible. Bangladesh must consider making a skill development capacity model to ensure graduates' skill development for sustainable development in the post-COVID-19 period. This study has showcased the importance of post-COVID-19 skill development for future employability for university graduates in Bangladesh, as well as their perception and needs. Based on the results, the recommendations made by this study will surely assist Bangladesh in formulating the skills development model to ensure

post-COVID-19 skills development and future employability for its university students, hence, ensuring sustainable development.

**Author Contributions:** Conceptualization, M.J.A. and K.O.; methodology, M.J.A. and S.R.B.I.; software, M.J.A. and S.R.B.I.; validation, K.O. and S.R.B.I.; formal analysis, M.J.A. and S.R.B.I.; investigation, M.J.A. and S.R.B.I.; resources, M.J.A. and S.R.B.I.; data curation, M.J.A. and S.R.B.I.; writing—original draft preparation, M.J.A. and S.R.B.I.; writing—review and editing, K.O.; visualization, M.J.A. and S.R.B.I.; supervision, M.J.A. and K.O.; project administration, M.J.A. and K.O.; funding acquisition, M.J.A. All authors have read and agreed to the published version of the manuscript.

**Funding:** This research was funded by the Japan Foundation, Tokyo. The JF Research Grant number (Ref. No. 10131980-002, FY 2021-2022).

**Institutional Review Board Statement:** The study was conducted in accordance with the Declaration of Helsinki, and approved by the Institutional Review Board (or Ethics Committee) of Department of Japanese Studies, University of Dhaka, Bangladesh (30 July 2021).

**Informed Consent Statement:** Participants in the research provided informed consent after being informed of the study's purpose.

**Data Availability Statement:** The research data can be obtained by contacting the corresponding author for more information. Due to the confidentiality of the information, it is not available to the public.

**Acknowledgments:** We want to acknowledge the staff of the Japan Foundation, Tokyo, for their spontaneous and generous support in conducting this joint research, without which the publication of this study would not be possible.

**Conflicts of Interest:** The authors declare no conflict of interest.

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
