# Peer review of "Importance of Skills Development for Ensuring Graduates Employability: The Case of Bangladesh"

_socsci, doi:10.3390/socsci11080360_

Round 1

Reviewer 1 Report

Thanks for the opportunity to review the manuscript titled: "Importance of Skills Development for Ensuring Graduates Employability: The Case of Bangladesh."

The topic of the article is interesting but there are some flaws that have to be improved.

 In my opinion, there is a lack of scientific research. 

Although the literature review is interesting, no research model is defined, nor are hypotheses or research questions formulated.

The methodology is also incomplete, as the data collection instruments used are not explained. We do not know which questions were used, how deep they were, or on which authors were based to formulate them.

Data processing is very poor, as it is descriptive statistics, which is not suitable for an investigation of this nature.

The results found are insufficient to reach the conclusions they present.

Reviewer 2 Report

1. The paper is addressing a key issue for developing countries and for the job market in general.

The author can indicate COVID 19 as an important contingency, but not the main cause of unemployment.

2. Skills development is a wide concept that can be further described in the literature review. Perspectives and analysis of HE institutions and corporate world can be included.

3. The choice of the methodology should be further explained

4. Answers of participants regarding career's aspirations could be further explored in the discussion part.

5. Limitations of the study should be mentioned

6. Additional references could be add:

- Clarke, M. 2017. “Rethinking Graduate Employability: the Role of Capital, Individual Attributes and Context.” Studies in Higher Education 43 (11): 1923–1937.

Hurrell, S. A. 2016. “Rethinking the Soft Skills Deficit Blame Game: Employers, Skills Withdrawal and the Reporting of Soft Skills Gaps.” Human Relations 69 (3): 605–628.

- Forrier, A., and L. Sels. 2003. “The Concept Employability: a Complex Mosaic.” International Journal of Human Resources Development and Management 3 (2): 102–124.

Fugate, M., A. J. Kinicki, and B. E. Ashforth. 2004. “Employability: A Psycho-Social Construct, its Dimensions, and Applications.” Journal of Vocational Behavior 65: 14–38.

Reviewer 3 Report

The study uses mixed-method research with survey, and in-depth interviews, various probabilistic and non-probabilistic-tic sample selection methods were used. There are total of 437 participants responded. The reviewer thinks it is a well-written paper. The literature survey in the Introduction is adequate, and it has a sound research methodology.

In order to provide more information, it is suggested that the author can make some more comparisons in the results section.

Please check the Author Contributions, Funding, Institutional Review Board Statement, Informed Consent Statement, Acknowledgments. Authors should provide correct information

Reviewer 4 Report

The study deals with the employability of graduates in the labour market through the example of Bangladesh. The topic is relevant and crucial to ensure that higher education output does not increase unemployment, but that fresh graduates and their skills can be used locally for the economic growth of the country. The topic should be of interest to the reading public. 

I have made the following suggestions for revising the study:

- In the introduction chapter it would be good to read about the importance of graduate employability in an international context. This is a topic that appears frequently in the literature. 

- In chapter 2.1, it would be good to see a graph showing the evolution of the unemployment rate in Banglaesh in the years before the COVID-19 epidemic and now, either plotted on a monthly or quarterly basis.

- The chapter describing the methodology is clear and perhaps too detailed in places (chapter 3.2). The authors have used low-statistical methods to present the survey results. It would be good to review the database and carry out more in-depth analyses, e.g. cross-tabulation analyses, if possible. 

-In the summary section it would have been good to read the authors' own policy proposals (linked to lines 445-449). Please think about what could be done. 

Reviewer 5 Report

The manuscript is devoted to an important issue of the skills development for ensuring graduates employability, espocially after pandemic and caused by this situation financial crisis. The subject matter discussed in the article is an interesting one, and timely in the context of the pandemic's just mitigated effects.  The results of the research carried out, are useful from the scientific point of view and interesting for the readers, although the topic discussed is not new. The statement is transparent, factually correct and well structured. The cited bibliography confirms the thesis put forward by the author. Numerous tabular statements are sufficiently commented. The presented final conclusions sufficiently relate to the thesis put forward by the author. The English language and style are fine; the minor text editing faults (e.g. no clear indication of data sources) do not significantly affect the quality of the presented issue.

Round 2

Reviewer 1 Report

Dear(s) author(s)

Thank you very much for considering my comments and suggestions, congratulations on this new version.

I think your article, in its current version, can be published.

Thanks again for your effort.